# Contamination of Substrate-Coating Interface Caused by Ion Etching

**Peter Panjan \*, Aljaž Drnovšek, Miha Čekada**  **and Matjaž Panjan**

Department of Thin Films and Surfaces, Jožef Stefan Institute, Jamova 39, 1000 Ljubljana, Slovenia;
aljaz.drnovsek@ijs.si (A.D.); miha.cekada@ijs.si (M.Č.); matjaz.panjan@ijs.si (M.P.)
\* Correspondence: peter.panjan@ijs.si

**Abstract:** In–situ cleaning of the substrate surface by ion etching is an integral part of all physical vapor deposition (PVD) processes. However, in industrial deposition systems, some side effects occur during the ion etching process that can cause re-contamination. For example, in a magnetron sputtering system with several sputter sources and with a substrate holder located centered between them, the ion etching causes the contamination of the unshielded target surfaces with the batching material. In the initial stage of deposition, this material is redeposited back on the substrate surface. The identification of the contamination layer at the substrate–coating interface is difficult because it contains both substrate and coating elements. To avoid this problem, we prepared a TiAlN double coating in two separate production batches on the same substrate. In such a double-layer TiAlN hard coating, the contamination layer, formed during the ion etching before the second deposition, is readily identifiable, and analysis of its chemical composition is easy. Contamination of the batching material was observed also on seed particles that caused the formation of nodular defects. We explain the origin of these particles and the mechanism of their transfer from the target surface to the substrate surface. By comparison of the same coating surface area after deposition of the first and second TiAlN layers, the changes in coating topography were analyzed. We also found that after the deposition of the second TiAlN coating, the surface roughness slightly decreased, which we explain by the planarization effect.

**Keywords:** TiAlN hard coating; unbalanced magnetron sputtering; ion etching; surface topography; growth defects



## 1. Introduction

The adhesion of PVD hard coatings to substrates is the deciding factor determining their performance and success in industrial applications. In general, PVD coating adhesion is affected by both substrate properties (e.g., composition, microstructure, roughness, thermal expansion coefficient) as well as by the deposition parameters (e.g., temperature, bias voltage, internal stresses, thickness). The key condition for good coating adhesion is a clean surface of the substrate. The next requirement is a strong chemical bonding between the substrate atoms and the depositing atoms that can be formed only if a sufficient number of nucleation sites are available on the substrate surface. Two additional conditions must be also fulfilled: the interface between the substrate and the coating must have low porosity and it must not contain brittle intermetallic phases [1]. Several approaches are used for improving the coating adhesion:

- Primary cleaning process: external mechanical and chemical pretreatment of the substrate surface prior to the insertion in the deposition chamber in order to roughly remove contaminants (such as grease, oxides).
- In situ pre-treatment (heating and ion etching) of the substrate before the coating deposition process in order to remove the contamination that has formed since the primary cleaning process had been performed. Ion bombardment also creates a

number of new nucleation sites for the coating atoms, which significantly improves chemical bonds between the substrate atoms and the condensing atoms. The ion etching also increases the roughness of the substrate surface and thus strengthens the coating–substrate interface.

- After substrate cleaning by ion etching, the coating adhesion strength can be further improved by deposition of a metallic interlayer, which is typically Cr or Ti. An interlayer minimizes internal stresses in the deposited coating and it can also dissolve oxides that remain on the substrate surface after ion etching [2].

A precondition for good PVD coating adhesion is therefore the cleanliness of the entire tool surface that should be free from oxides and other undesirable contaminants. Otherwise, if the contaminant layer is not removed completely, then there is a high likelihood of coating delamination.

The first report about the cleaning of substrate surfaces by ion etching is from 1955 when Farnsworth et al. reported the use of ion etching with $Ar^+$ to prepare ultra-clean surfaces for low-energy electron diffraction studies [3]. Later, cleaning by ion etching was used by D. M. Mattox as a part of the ion plating deposition technique, first introduced by the author [4]. Today, in-situ cleaning of the substrate surface by ion etching is routinely used in all deposition systems for the preparation of PVD coatings. The cleaning procedure is performed by low-energy bombardment with inert gas or metal ions, extracted from glow discharge plasma, cathodic arc, or high-power impulse magnetron sputtering (HIPIMS) discharge.

Cleaning by ion etching is most often performed in Ar (or Ar + Kr) glow discharge plasma, where the gas ions are accelerated to several hundred eV by a DC or RF bias voltage on the conductive or non-conductive substrates, respectively [5,6]. The disadvantages of ion etching by inert ions are its small etching rate of oxides due to the low plasma density and the accumulation of insoluble inert gas atoms at the substrate–coating interface [7]. Namely, during sputter cleaning, the argon ions can be incorporated (up to several atomic percent) into the near-surface region of the substrate. Implanted ions of inert gas can cause tensile stresses and/or form an amorphous zone in the substrate surface region. During coating deposition or under exploitation of coated components, the incorporated gas atoms diffuse and agglomerate into bubbles, which introduce porosity and weakening of the interface [8]. To prevent the incorporation of gas atoms, ion etching should be performed at an elevated temperature (>300 °C) or the substrate should be annealed before film deposition.

An alternative approach to argon ion etching is metal ion etching with a cathodic arc discharge serving as an ion source. Cathodic arc discharge produces a highly ionized metal flux without using a process gas. Bombardment with metal ions is known to provide not only a cleaner substrate surface but to also produce very thin (a few nm thick) implantation zones that can promote localized epitaxial growth of the coating. The disadvantage is the contamination of the substrate surface with the macroparticles (droplets) of the cathode material. Because their bonding with the substrate material is poor, they reduce the adhesion of the coating. There is also often a risk (especially for small tools) that too intensive etching leads to local substrate overheating and consequently softening effects [9,10].

Contamination of the substrate surface with droplets can be avoided by using a high-power impulse magnetron sputtering (HIPIMS) system [11–13]. Highly ionized HIPIMS plasma contains a mixture of single-, double-, and sometimes even higher charged metal and argon ions. The bombardment of the substrate surface with metal ions causes the formation of a shallow metal implantation zone, while the crystallinity of the substrate surface is not destroyed. This improves the bond between the substrate material and the subsequently deposited coating and thus enhanced adhesion. The HIPIMS etching of the coatings exhibits a defect-free smooth surface without any droplets. Therefore, PVD coatings deposited after HIPIMS pretreatment exhibit a superior adhesion in comparison to pretreatments performed in argon glow discharge and cathodic vacuum arc environments.

In the literature, most data refer to the etching of flat substrates in experimental deposition systems. Much more demanding is the ion etching of substrates (tools) in

industrial deposition systems equipped with more sputtering sources and several substrate holders (towers). There are only a few studies that address the problem of etching tools in industrial systems for deposition of PVD hard coatings, where a typical job-coating batch is filled with different types of tools, which can vary considerably in size and geometry [14,15]. The electric field between the plasma and the tool attracts ions from the plasma. Since the electric fields are more localized at the sharp edges, rounded surfaces and edges are subjected to different ion flux densities as compared to a flat surface. Therefore, highly non-uniform ion etching, not only on different tools within the batch but also on different areas of the same tool, can be expected Therefore, it is almost impossible to predict the etching efficiency within a batch. Most often, information on the efficiency of etching in industrial systems is acquired empirically. Avoiding this problem requires a very careful design of the substrate table and holders. In addition, a well thought-out loading of tools by skilled operators is needed.

A big challenge in sputter cleaning of tools in a typical job-coating batch is how to achieve uniform ion current density over the entire tool surface. A more uniform etching can be achieved by using multiple rotations of tools. During ion etching and deposition processes, the smaller tools, such as drills, milling cutters, taps, and inserts are exposed to a triple planetary rotation [16]. They are mounted on several rotating satellites and depending on tool size, each satellite has one or several levels. We must also take into account that during ion etching, not only the tool surface, but also structural components (such as load fixtures) of the deposition system are sputtered and can cause additional contamination of tool surfaces.

In our recently published studies, we described the topographic changes of the substrate surface during ion etching [5,6]. We also pointed out that the etching efficiency depends on batching configuration and etching parameters. In such a system, there is a high probability that re-deposition and cross-contamination of the substrates will occur. Namely, in areas where the intensity of ion bombardment is low, contaminates may accumulate due to the re-deposition of sputtered material from areas where the intensity is high. The next problem is the material already deposited on the substrate holder, which is sputtered off and can be redeposited on the new batch of tools. All these phenomena, which can reduce the usefulness of cleaning by ion etching, must be properly taken into account to minimize their influence. In this paper, therefore, we focus on the cross-contamination and re-deposition problems in industrial deposition systems equipped with four rectangular unbalanced magnetron sources.

## 2. Materials and Methods

The industrial magnetron sputtering system (modified CC800/7, CemeCon, Würselen, Germany) was used for the deposition of the single layer TiAlN and multilayer TiAlN/CrN hard coatings. All experiments were performed in production batches. In a typical batch, different tools made of cemented carbide, high-speed steel (HSS), or cold work tool steel, were loaded on specially designed substrate holding fixtures. The TiAlN coating composition was Ti 23 at.%, Al 27 at.%, and N 50 at.%. The test substrates in the form of discs were made of powder metallurgical (PM) ASP30 tool steel (Uddeholm, Hagfos, Sweden), cold work tool steel AISI D2 (Ravne steel factory, Ravne, Slovenia), and cemented carbide (HM). All substrates were first ground and polished to a mean roughness of $R_a$ = 10 nm. Prior to the coating process, they were cleaned in detergents (alkaline cleaning agents, pH $\approx$ 11) and ultrasound, rinsed in deionized water, and dried in hot air. In the vacuum chamber, they were first heated to about 450 °C and then in-situ cleaned by radio frequency (RF) ion etching in an argon atmosphere. The RF power and the etching time were 2000 W and 90 min, respectively. The details of the ion etching and deposition process are described in a recently published article [17]. The TiAlN coating thickness was around 4 μm as measured using a ball crater technique. The total operating pressure was maintained at 0.75 Pa, with the flow rates of argon, nitrogen and krypton being 100, 160, and 80 mL/min, respectively. A DC bias of −100 V was applied to the substrates. The duration of the deposition process

was 135 min. After this time, the deposition process was interrupted for an intermediate ion etching (for 60 min under the same conditions as during substrate cleaning). This was followed by an additional deposition of a TiAlN coating (deposition time was 30 min). The intermediate etching creates new nucleation sites for the subsequently deposited nitride coating, resulting in a fine-grained and less porous microstructure of the top layer [18].

After the deposition, the samples were taken out of the vacuum chamber and the TiAlN coating surface was analyzed by 3D stylus profilometry and SEM microscopy to verify the presence of growth defects and other morphological features. The surface topography characterization of the coated and uncoated substrate was carried out using scanning electron microscopy (SEM, JEOL JSM-7600F, Tokyo, Japan) and 3D stylus profilometry (Bruker Dektak XT, Billerica, MA, USA). The evaluation area of profilometer was 1 mm$^2$ with a resolution of 0.2 μm in x and 1 μm in the y direction, while the vertical resolution was around 5 nm. After these investigations, the coated samples were ultrasonically cleaned and then put again into the deposition system and coated with the second TiAlN layer prepared in the same manner as the first one. This means that before the second deposition, the coated substrates were exposed to the intensive ion etching again. After the second deposition, the surface topography and the microstructure of TiAlN double layer were analyzed again.

Two Cr targets and two mosaic TiAl targets were used for the deposition of TiAlN/CrN multilayer coating, where the Cr targets were positioned on one side of the vacuum chamber and TiAl targets on the other. The samples were mounted in a one-fold rotation mode. The total thickness of the coating deposited on the cemented carbide substrate was about 12 μm, while the thicknesses of individual TiAlN and CrN layers were around 50 nm and 70 nm, respectively. The Zeiss Axio CSM 700 confocal optical microscope (Zeiss, Jena, Germany) was used to observe the ground section of the coating.

The microstructure and the coating morphology were studied using fracture cross-sections examined using a field emission scanning electron microscope (FEI Helios Nanolab 650i, Amsterdam, The Netherlands). Cross-sections were also prepared by the focused ion beam technique (FIB) using an FIB source integrated into the FEI SEM scanning electron microscope. SEM images were recorded using the ion beam and the electron beam. EDS mapping was carried out by using the Oxford Instruments system attached to the SEM.

Scanning transmission electron microscopy (STEM) was performed using a JEOL ARM 200 CF (Jeol Ltd., Tokyo, Japan) operated at 200 kV and a Jeol Centurio 100 mm$^2$ SDD EDXS system. All TEM images and analyses were obtained on cross-sections. The diameter of the electron beam was around 0.1 nm, but due to beam broadening and taking into account sample thickness, chemical composition, and density, the lateral resolution was around 10 nm. The FIB lift-out method was used for the preparation of TEM specimens.

## 3. Results and Discussion

### 3.1. Surface Topography of Single and Double Layer TiAlN Coatings

After deposition of the first TiAlN coating, the designated surface area was examined by 3D stylus profilometry and SEM microscopy. In SEM images at low magnification (Figures 1 and 2), growth defects of various shapes and sizes can be observed. Protrusions at sites of carbides in the steel substrate that were formed during ion etching of the substrate are visible. In SEM images at high magnification of the area between various protrusions, we can see that the coating surface exhibited a faceted domain-like morphology related to its columnar growth (Figure 1). Selected topographical features of the TiAlN coating were pinpointed. After these analyses, the coated samples were put again into the deposition system and coated with the second TiAlN coating prepared by the same procedure as the first one (heating, ion etching, deposition of TiAlN coating). The optical image of the ball crater through such double-layer TiAlN coating is shown in Figure 3a. The bright rings belong to the TiAlN layers that were formed after the intermediate ion etching of the individual coatings at about three-quarters of the deposition time [18]. Such ion etching causes re-sputtering of otherwise rounded column tops and the formation of new sites for

nucleation of the subsequently deposited coating. These processes change the topography and microstructure of the layer that grows after the interruption. Figure 3b shows the SEM image of the FIB cross-section of TiAlN double-layer deposited on the PM ASP30 tool steel substrate. The interface between the first and the second TiAlN coatings is clearly visible. The EDS analysis showed that tungsten, iron, and chromium are present at the interface. The reasons for such contamination are explained later.

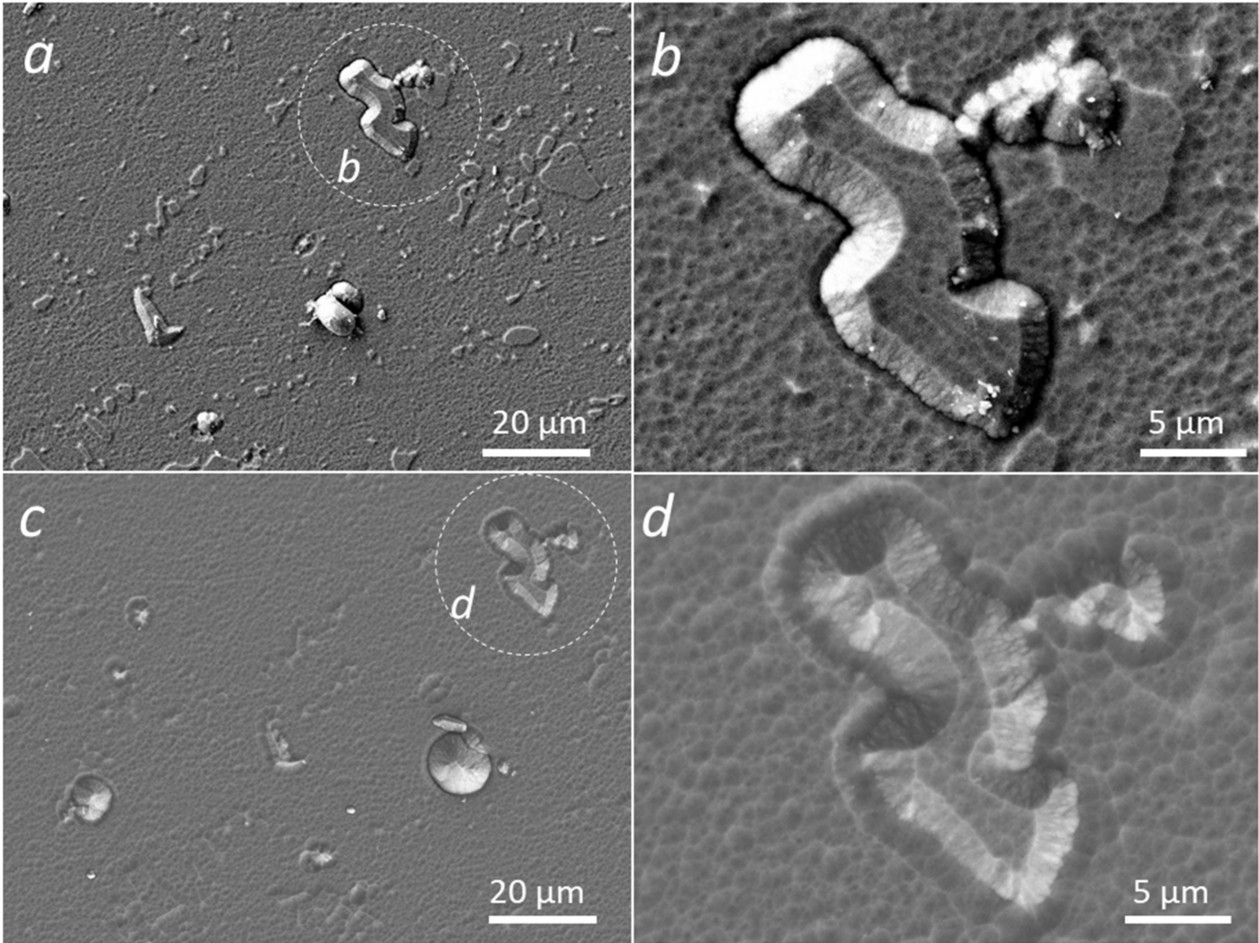

**Figure 1.** SEM micrographs at a lower (**a**,**c**) and higher magnification (**b**,**d**) of the same surface area of the single (**a**,**b**) and double layer TiAlN (**c**,**d**) coatings deposited on D2 substrate.

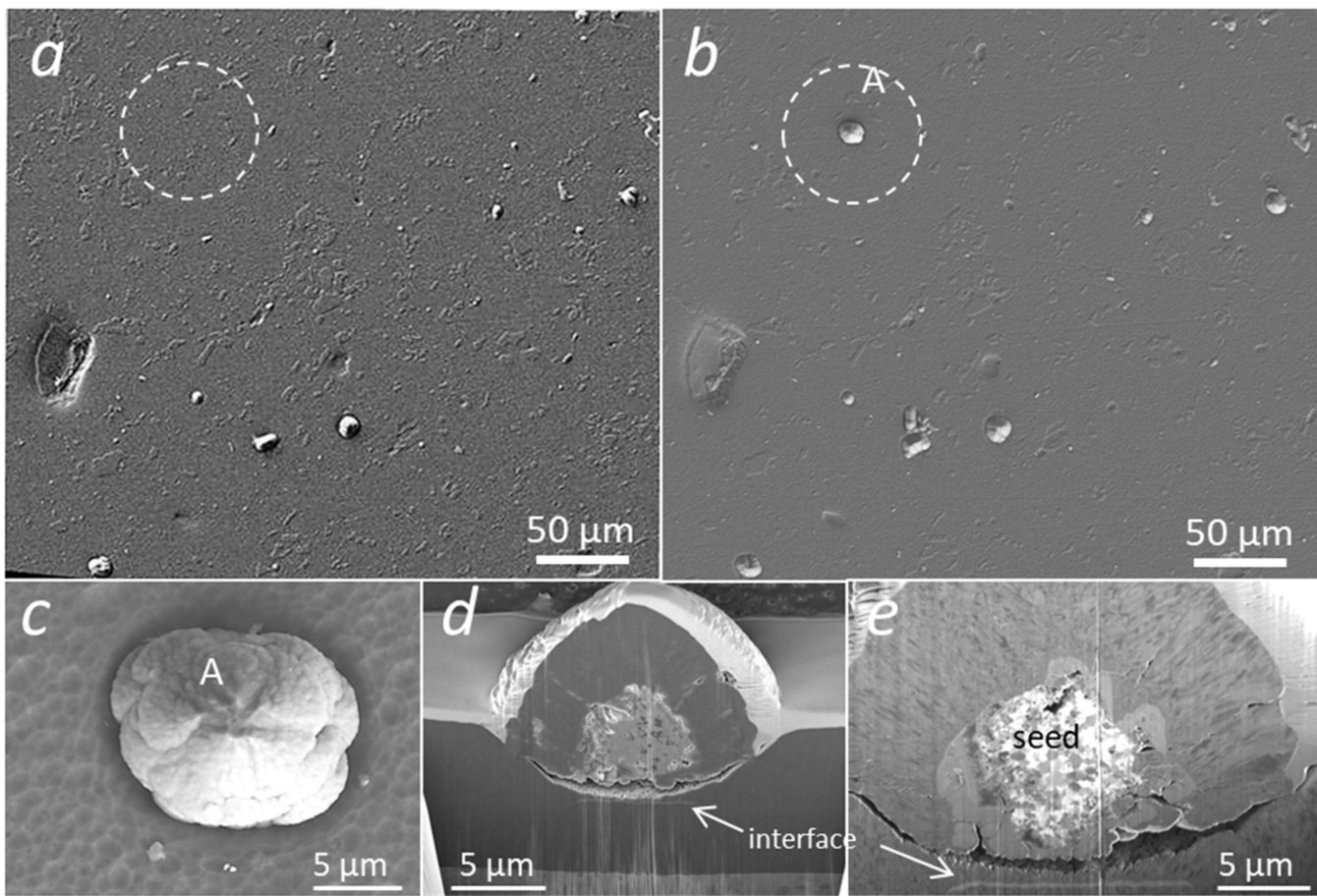

**Figure 2.** SEM micrographs at a low magnification of the same surface area of the single (**a**) and double layer (**b**) TiAlN coatings deposited on the D2 substrate. Plain-view SEM micrograph (**c**) and FIB secondary electron images (**d,e**) of cross-sections of a selected nodular defect (A) formed during deposition of the second TiAlN coating.

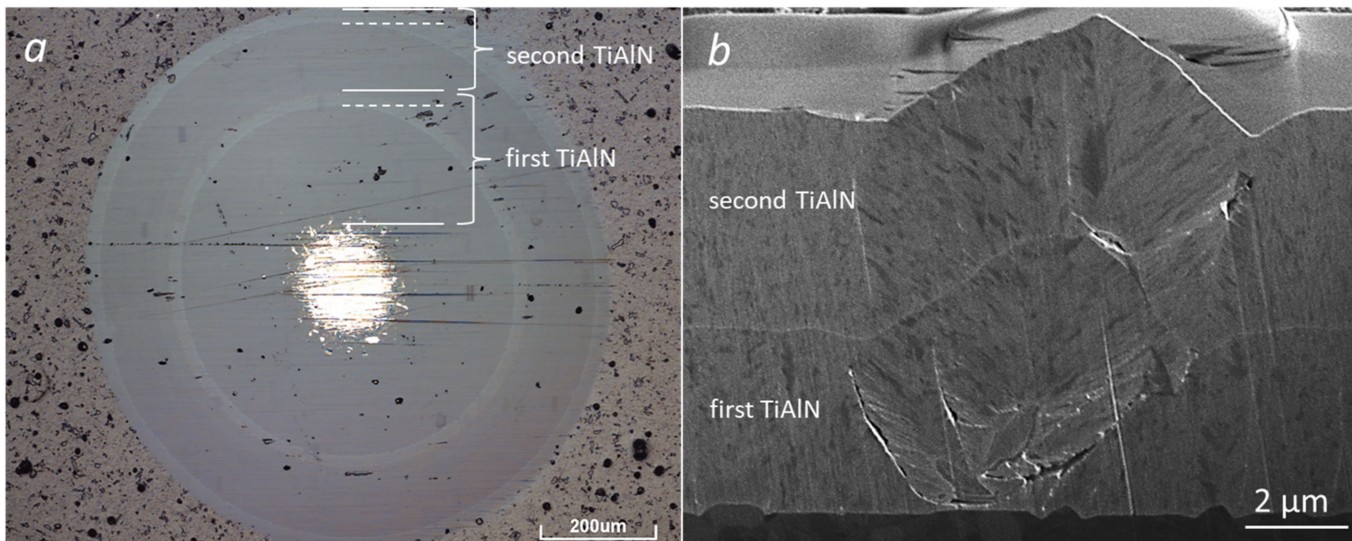

**Figure 3.** (**a**) Ball crater through a double layer TiAlN coating. The bright rings belong to the TiAlN layer formed after the intermediate ion etching of the coating. (**b**) SEM micrograph of an FIB cross-section of the TiAlN double layer deposited on the PM ASP30 tool steel substrate. The interface between the first and the second TiAlN coatings is visible as a continuous bright line.

The topography of the double-layer TiAlN coating surface was examined again by following the same surface area. SEM images in Figures 1 and 2 show the comparison of the size and shape of selected growth defects after the first and after the second deposition of TiAlN coating. We found that after the deposition of the second TiAlN coating, most nodular defects remained on the coating surface, some of them converted into craters, while some new nodular defects also appeared. The conversion of nodular defects into craters most likely occurs due to the high thermal load during ion etching. The resulting thermal stresses can cause the detachment of some weakly bonded nodular defects [19,20]. We can also observe that the surface of the double layer TiAlN coating is smoother in comparison with the single one. This planarization phenomenon is due to ion etching before the deposition of the second TiAlN coating. Namely, during ion etching, the side surface of all protrusions etches faster than the flat coating surface because the sputtering yield at the normal incidence of ions is much smaller than at a high incidence angle [6]. This effect leads to the shrinking of all protrusions and even elimination of the smaller ones [5]. This is the reason why the protrusions formed at carbide sites after the deposition of the first TiAlN coating on the ASP tool steel substrate almost disappear after the second deposition of the TiAlN coating (Figure 4).

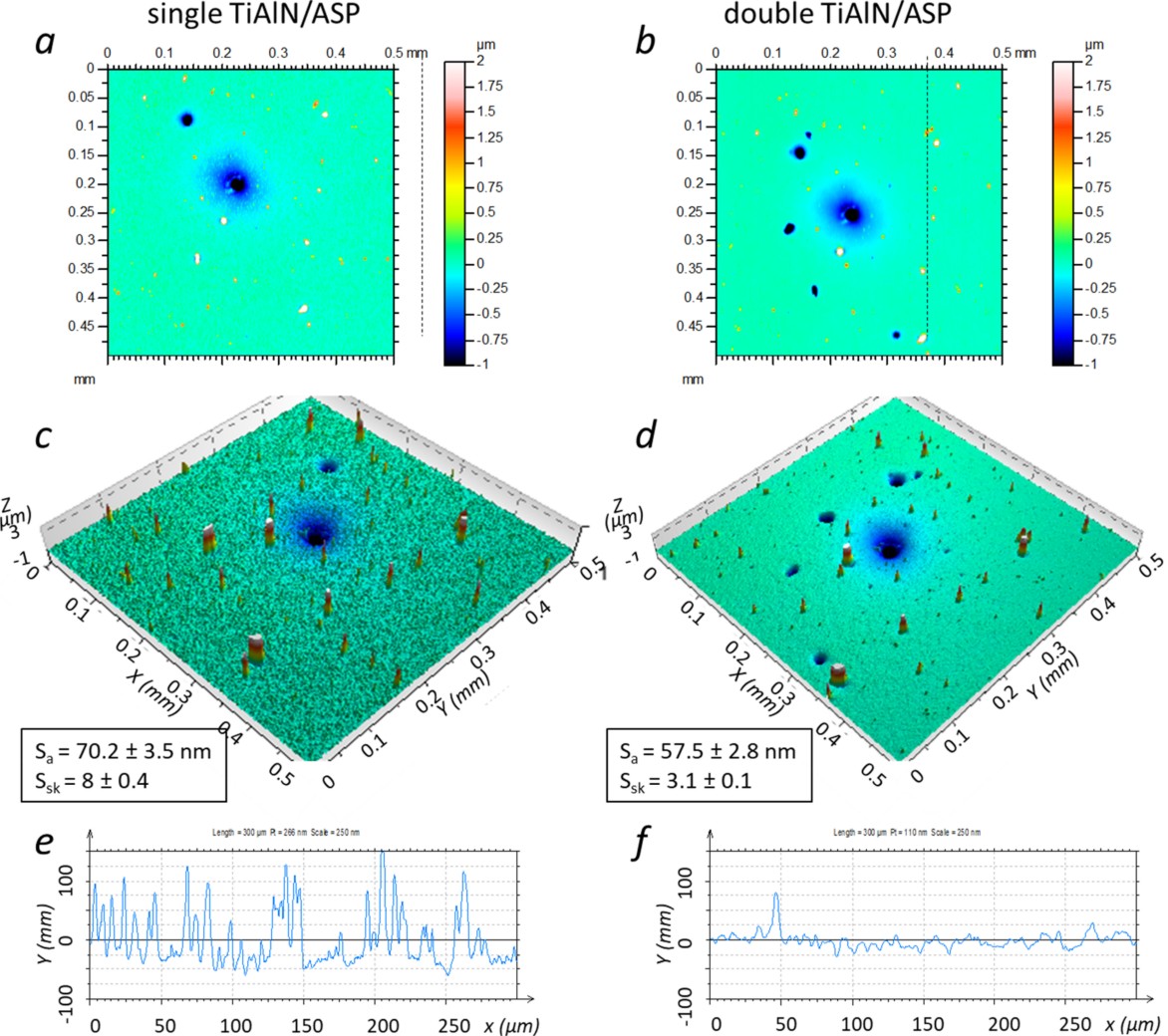

**Figure 4.** Top view (**a**,**b**) and 3D profilometer images (**c**,**d**) of the same surface area of single and double layer TiAlN coatings deposited on the ASP tool steel substrate. Roughness parameters ($S_a$, $S_{sk}$) of the substrate, after deposition of the first and the second deposition of TiAlN coatings, are given in the frames. The line profiles (**e**,**f**) were recorded on the defect-free areas of 3D profile images (**c**,**d**).

However, if the TiAlN coating is deposited on a D2 substrate, where the protrusions at carbide sites are much larger (Figure 5), they do not disappear completely. The effect of ion etching on protrusions at carbide sites is clearly seen from the roughness line profiles. They were recorded on the defect-free areas of the TiAlN coating (Figures 4 and 5). All peaks on these profiles belong to protrusions formed at carbide inclusions. It is evident that after ion etching before the deposition of the second TiAlN coating on the ASP tool steel substrate these peaks disappear, while they are not eliminated completely if D2 tool steel is used for the substrate. The planarization effect due to preferential ion etching is reflected in the reduction of surface roughness $S_a$ after deposition of the second layer of TiAlN. The surface skewness $S_{sk}$, which is a measure of the asymmetry of the surface profile from the surface mean line, is also reduced. Due to carbide protrusions formed during the polishing of bare substrates, the value of the parameter $S_{sk}$ was positive for both bare ASP and D2 tool steel substrates (Table 1). The $S_{sk}$ roughness parameter for bare ASP substrate was much higher than that for bare D2 substrate. The reason for this phenomenon is that during mechanical pretreatment of D2 substrates, where large carbide inclusions are present, a part of protruded carbides is torn out due to large shear stress leaving a pit in the substrate. The formation of such pits causes the reduction of the $S_{sk}$ roughness parameter. It remained positive but smaller after the deposition of the first TiAlN. Changes in the roughness parameters are caused both by ion etching of the bare steel substrate as well as by deposition of the TiAlN coating. Due to the inhomogeneity of the D2 and ASP tool materials and the consequent different etching rates of various phases and grains with different orientations, both shallow depressions and protrusions formed. This is reflected in both higher roughness $S_a$ and lower skewness $S_{sk}$ (Table 1). After deposition, the skewness parameter increased due to the formation of nodular defects. Shrinking of all protrusions during ion etching, performed before the deposition of the second TiAlN coating (Figures 1 and 2), is reflected in lower values of roughness parameters, both surface roughness $S_a$ and surface skewness $S_{sk}$. For the reasons given above, the reduction in both roughness parameters is more pronounced in the layers applied to the ASP substrate.

**Table 1.** Roughness parameters ($S_a$, $S_{sk}$) of ASP and D2 substrates, after polishing, after deposition of the first TiAlN coating and after deposition of the second TiAlN coating. The scanning area was 0.5 mm × 0.5 mm.

| Substrate | $S_a$ (nm) | $S_{sk}$ |
|---|---|---|
| bare ASP | $11.8 \pm 0.6$ | $10 \pm 0.5$ |
| ASP + TiAlN | $70.2 \pm 3.5$ | $8 \pm 0.4$ |
| ASP + TiAlN + TiAlN | $57.5 \pm 2.8$ | $3.1 \pm 0.1$ |
| bare D2 | $11.3 \pm 0.6$ | $2.9 \pm 0.1$ |
| D2 + TiAlN | $83.1 \pm 4.1$ | $7.5 \pm 0.4$ |
| D2 + TiAlN + TiAlN | $82.6 \pm 4.1$ | $3.2 \pm 0.1$ |

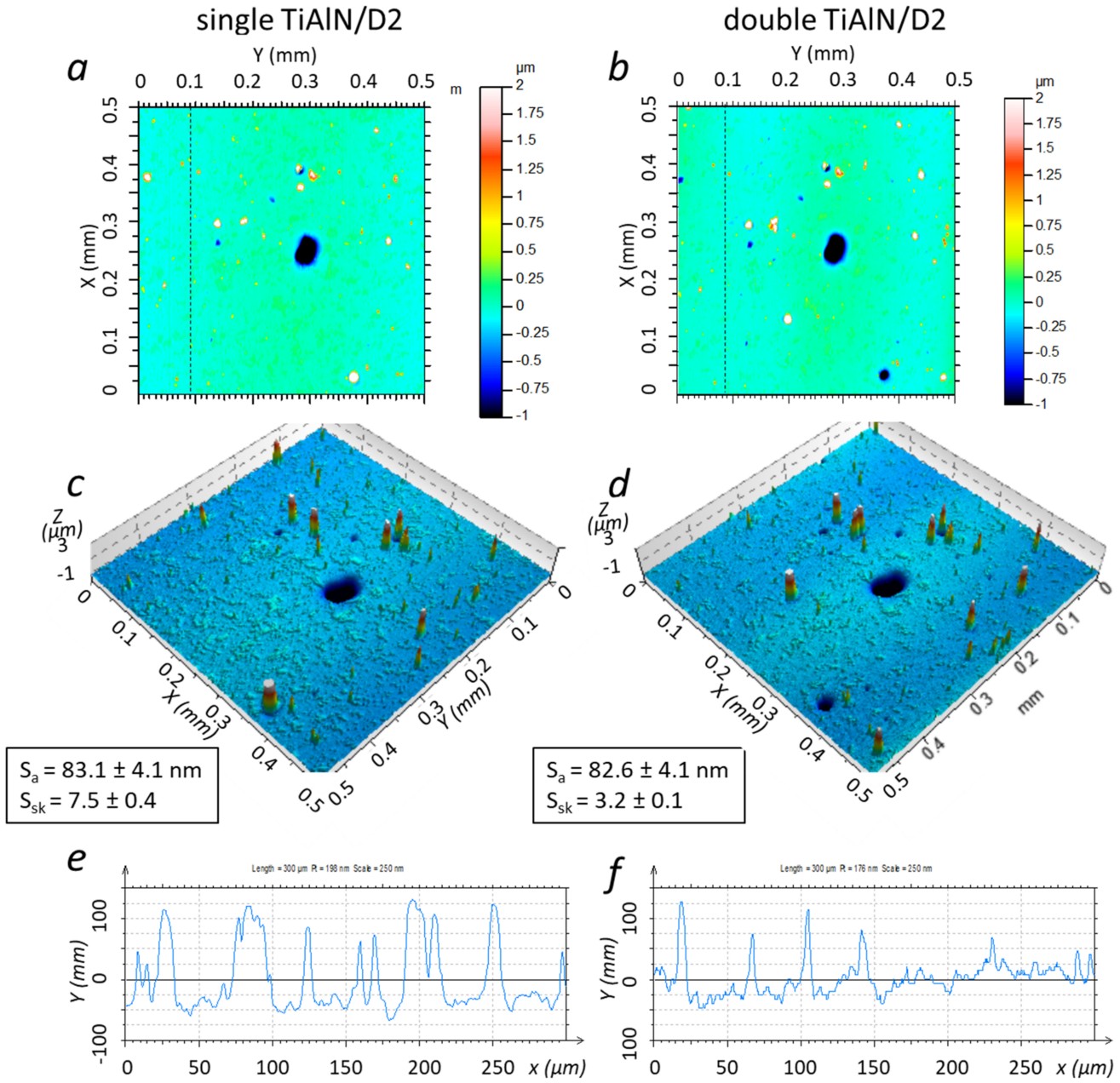

**Figure 5.** Top view (**a**,**b**) and 3D profilometer images (**c**,**d**) of the same surface area of single and double layer TiAlN coatings deposited on the D2 substrate. Roughness parameters ($S_a$, $S_{sk}$) of the substrate, after deposition of the first and the second deposition of TiAlN coatings, are given in the frames. The line profiles (**e**,**f**) were recorded on the defect-free areas of 3D profile images (**c**,**d**).

Additional phenomena are well-defined trenches or depressions occurring in the boundary region of sputtered nodular defects (Figure 6). Trenching is caused due to enhanced erosion near the sidewall of nodular defects (see Ref. [6]).

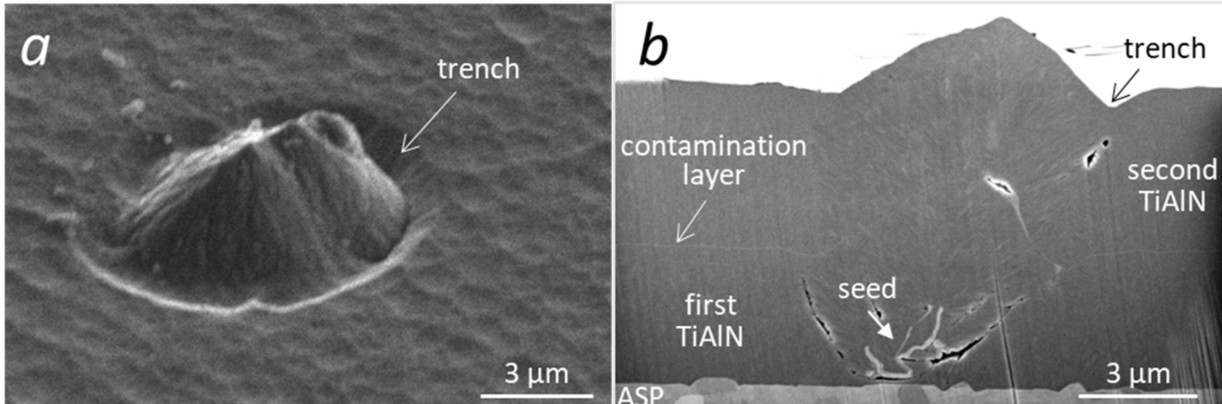

**Figure 6.** Plain-view SEM micrograph (**a**) and SEM micrograph of the FIB cross-section (**b**) of a nodule on the TiAlN coating. A trench around the nodular defect as well as a contamination layer on the seed particle is clearly visible.

### 3.2. Contamination of the Interfacial Region in Double-Layer TiAlN Coating

The microstructure and compositional depth profile of the resulting double-layer TiAlN coating were investigated by cross-sectional SEM (Figure 7) and STEM microscopy (Figure 8). High-resolution bright filed imaging indicated that the contamination interlayer formed with a well-defined thickness (around 5 nm) (Figure 8a). The interface between the substrate and the film is smooth and abrupt. The STEM-EDX line profiles show that it is composed mostly of iron, chromium, and tungsten (Figure 8b). The most likely origin of these metal elements is the batching material, which was in our case HSS and cemented carbide cutting tools. The detection of batching material in the interfacial region is probably due to the substrate ion etching before the coating growth. In an ion etching process, the sputtered atoms of the batching material are deposited on all surfaces in the vacuum chamber, including the target surfaces that are not protected by a shutter (Figure 9). Deposited material on the target surface subsequently re-deposited back on the substrates during the initial stages of the coating growth. In addition to the batching material, the interfacial region was also contaminated with the coating material, which was deposited on the substrate fixturing components in the previous batch.

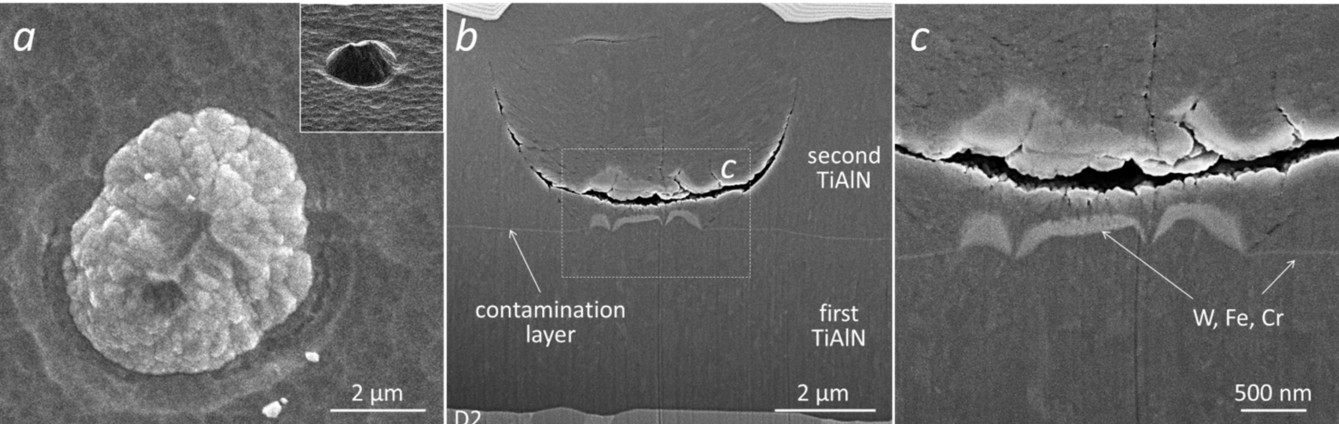

**Figure 7.** Plain-view SEM image (**a**) of a nodular defect formed in the initial stage of deposition of the second TiAl coating; (**b**,**c**) FIB cross-section of TiAlN double layer coating clearly shows the contamination layer between both TiAlN coatings and under the nodular defect.

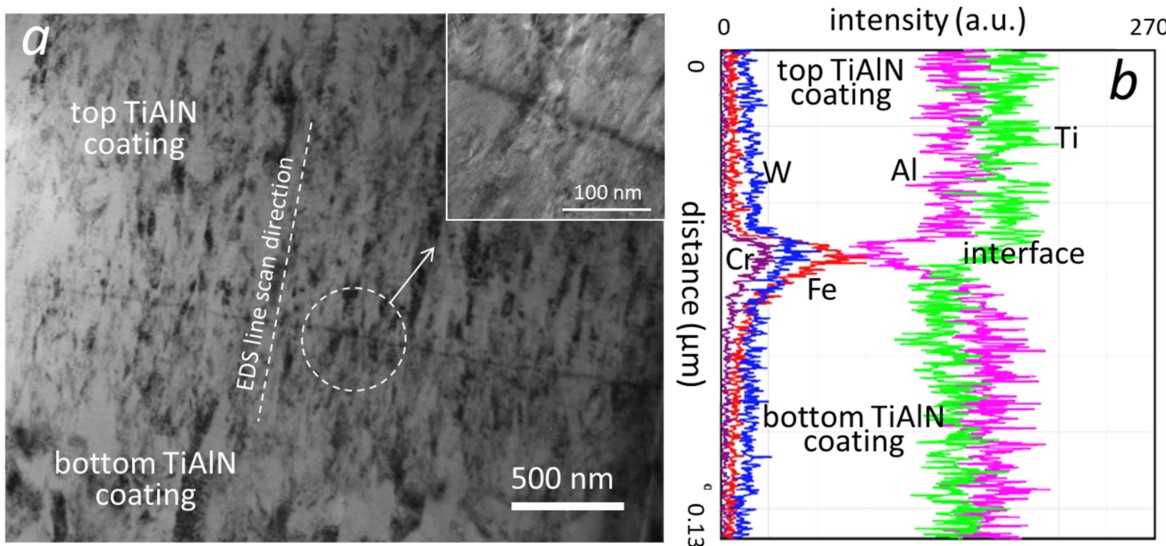

**Figure 8.** (**a**) Cross-sectional STEM micrograph and (**b**) STEM-EDX compositional profile of the interface region.

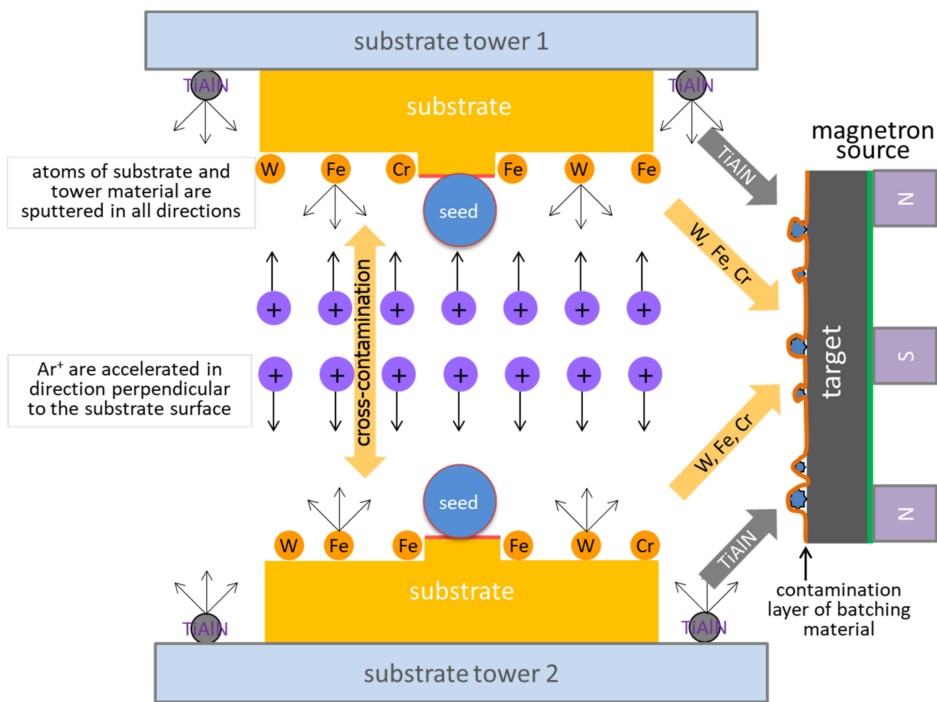

**Figure 9.** Schematic of the ion etching process and subsequent contamination of substrates and target surface with the batching material.

The interfacial region between the substrate and the first TiAlN coating is also contaminated in the same way. However, at this interface, it is very difficult to distinguish between substrate-related elements originating from contamination and those from the substrate itself [21,22]. The contamination of the target with the residual products from the etching process can be prevented by a movable shutter located close to the target, which collects the sputtered species and enables the pre-sputtering of targets. However, the majority of industrial deposition systems are usually not equipped with movable shutters because the complicated installation of such a system reduces the economy of the deposition process.

During the ion etching process, cross-contamination also occurs, i.e., when the sputtered atoms removed from one substrate tower contaminate the other one and vice versa.

An effective sputter cleaning is possible only if the mean number of sputtered atoms per unit surface area is higher than the number of re-sputtered atoms (and other molecules from the residual gas) hitting and sticking on the substrate surface. For the majority of industrial deposition systems, this requirement is fulfilled. However, contamination may occur on the substrate surfaces shaded beneath foreign particles (Figure 7b,c). These areas are not exposed to ion etching because the ions are directional perpendicular to the substrate surface. On the other side, the same surface is covered with atoms of the batching material, which can arrive from areas where the intensity is high. Therefore, the thickness of the contamination layer at sites of foreign particles is significantly larger (see Figure 7c).

### 3.3. SEM Images of Broken Nodular Defects

During or after the deposition, some nodular defects are broken off due to the high residual stress in the coating. The backscattered electron (BSE) images of the broken nodular defects are shown in Figure 10. In these images, we can see that the seed particles are surrounded by a thin contamination layer. EDS analysis shows that this layer has a similar composition as the contamination layer formed at the interface between both TiAlN coatings. Based on this fact, we can conclude that the origin of metal elements (W, Fe, Cr) is the same in both cases. However, the key question is the origin of these seed particles that form growth defects. In the PVD deposition systems with magnetron sputtering sources, the formation of such particles (flakes) can be caused by (a) flaking of cones formed in the target racetrack, (b) flaking of the redeposited nodules from the target surface, and (c) by arcing [6,23]. Due to the electrostatic self-repulsion effect, a part of the flakes reach the substrate surface, where they are built into the growing coating.

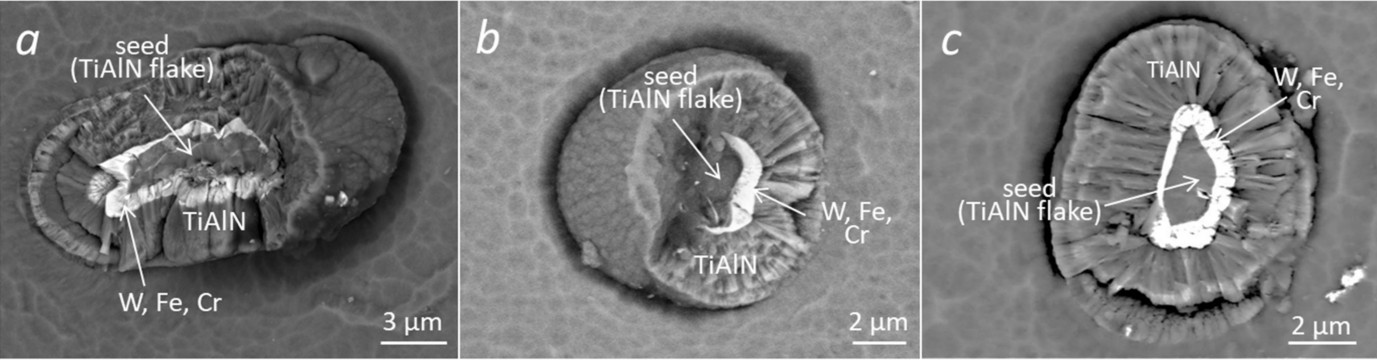

**Figure 10.** BSE images of broken nodular defects (**a**–**c**) show seed particles with an envelope that is composed of mostly of iron, chromium, and tungsten.

A still more intensive source of seed particles is in the target area outside of the racetrack. During the sputtering process, the target material is also re-deposited from the racetrack to the perimeter. Due to the target poisoning effect and low plasma density, sputtering does not take place in the perimeter region. High internal stress causes the re-deposited material to grow in the form of weakly bonded filaments [24,25]. Filaments formed near the racetrack eventually cross the high-density plasma region where they are heated. This causes their fracture into smaller fragments that are charged negatively accelerating them away from the target due to the electrostatic self-repulsion effect. A part of them reaches the substrate surface and during the deposition process, they cause the formation of nodules. In addition to the particles formed by the mechanism described above, there are also other particles on the target surface. These are, for example, flakes which have spalled off from the vacuum chamber components (e.g., shields, substrate fixture system) during the heating process or during the ion etching. There are also dust particles left on the surface of the target during the loading of the batching material since complete removal of these particles before deposition by blowing and wiping is not possible. During ion etching, these particles are also covered with a contaminant layer. At

the beginning of the deposition process, an electric charge accumulates on these weakly bonded particles and electrostatic forces cause self-repulsion of them towards the substrate.

### 3.4. Depth Distribution of Nodular Defects

In the previous section, we found that many seed particles that caused the formation of nodular defects originated from the target surface, meaning that nodular defects start to form mostly at the beginning of the deposition process. However, the question is whether this mechanism of nodular defect formation is predominant. To answer this question, we tried to determine the depth distribution of nodular defects from the ground section prepared by the ball-cratering technique (Calotest). Because nodular defects on the ground section of a single TiAlN coating are not visible, we performed the analysis on a TiAlN/CrN multilayer structure deposited in the same system (Figure 11a).

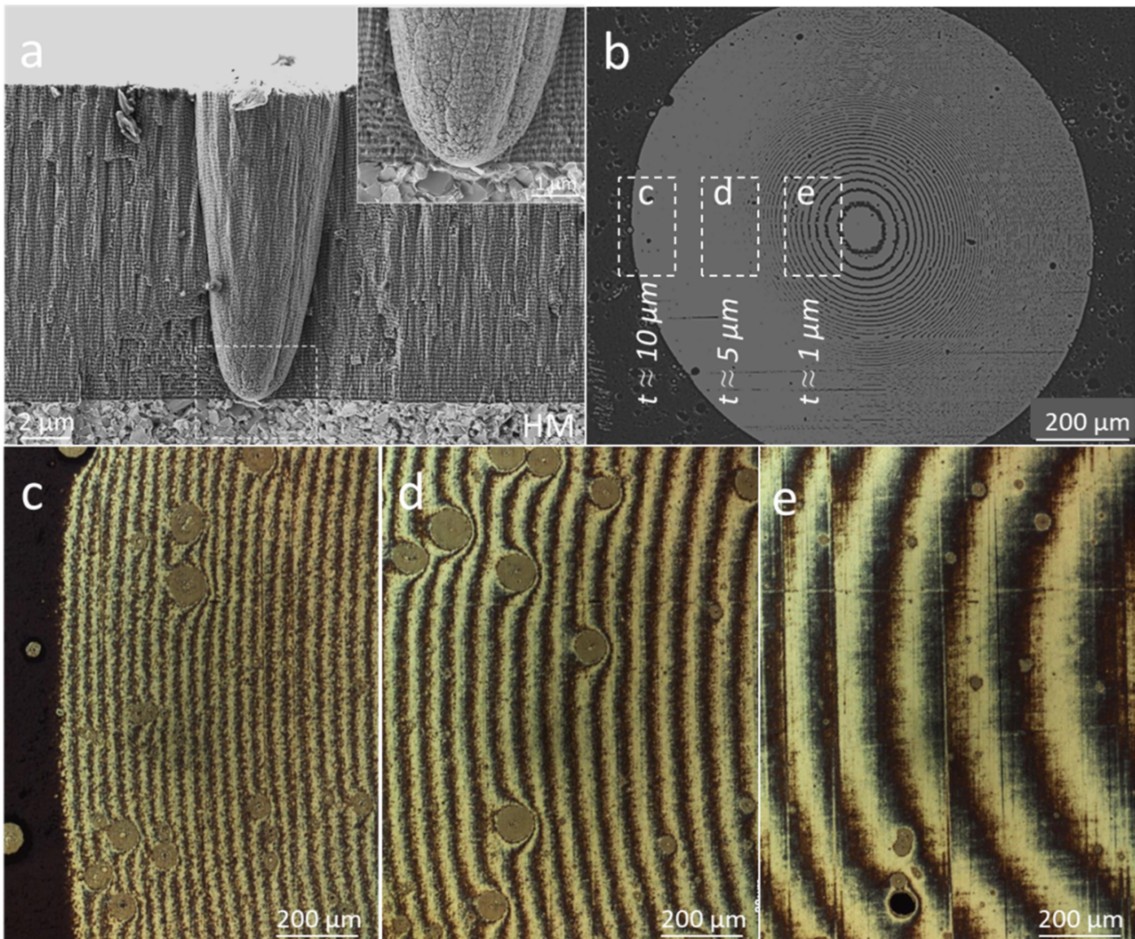

**Figure 11.** Fracture cross-sectional SEM micrograph of the TiAlN/CrN multilayer coating on HM substrate (**a**) and OM images of the ground section of TiAlN/CrN multilayer coating at low (**b**) and high magnifications (**c**–**e**). The density of nodular defects at different depths (frames (**c**–**e**)) is similar (20–30 defects/mm$^2$). Due to a parabolic shape, the nodule cross-section increases with coating thickness.

On a low-angle cross-section of such coating, the layer contours reveal the positions of defects (Figure 11c–e). Thus, we can count how many nodular defects are located at a certain coating depth. We found that the concentration of defects in the middle of the coating and close to the top surface is comparable with the concentration close to the substrate–coating interface. The more or less uniform distribution of nodular defects on the ground section of the TiAlN/CrN coating means that most nodules started to grow at the substrate–coating interface. Namely, we have to consider, that all nodular defects

formed at the substrate–coating interface extend through the entire coating. Due to the parabolic shape, the cross-section of nodular defects close to the substrate is much smaller in comparison with that at the coating surface.

　　　FIB cross-sectioning of selected nodular defects on the ground section showed that all of them were formed on the substrate surface (Figure 12). This could mean that the mechanism of defect formation described in the previous sections is most likely predominant. This is also confirmed by the fact that a thin layer of contamination from heavy elements is clearly visible on some seeds (Figure 12c–f).

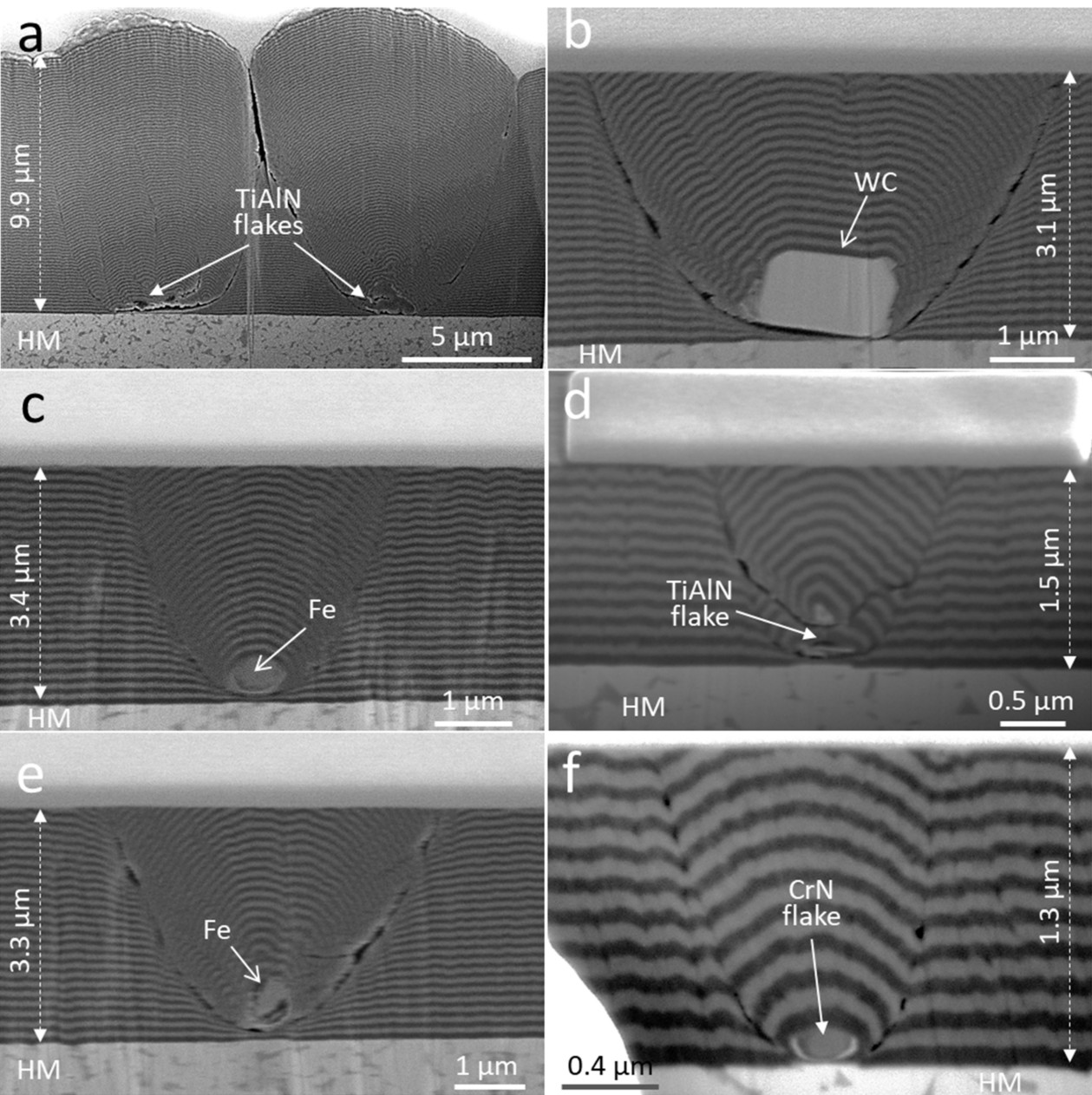

**Figure 12.** FIB cross-sectioning of selected nodular defects on the ground section of the TiAlN/CrN multilayer coating at different depths (**a**–**f**). SEM images were obtained with a BSE detector to emphasize the dissimilarity between TiAlN and CrN layers. Due to its high atomic number, the electron backscattering on CrN is more intense and is therefore displayed as layers with brighter contrast.

### 4. Conclusions

In this paper, we focused on the problem of target surface contamination in an industrial magnetron sputtering deposition system. Such contamination with the residual products from the etching process can be prevented by a movable shutter located close to the targets, but in order achieve a reasonable economics of the deposition process, complicated installations (including shielding and shuttering) are usually avoided. In this paper, all experiments were performed on samples prepared in the production batches. In a typical batch, several types of tools are loaded and different fixtures are used, which are designed for different tooling arrangements. The contamination layer at the interface between the substrate and the coating is difficult to identify because it contains both substrate elements (batching material) and coating elements (the coating material deposited on the substrate fixturing components in the previous batch) To avoid this problem, we deposited a TiAlN double coating in two separate production batches on the same substrate. Using this approach, the contamination layer between the two TiAlN coatings was easier to identify and determine its composition and thickness. We found that in such layer metal elements (W, Fe, and Cr) are present, which originate from the batching material (steel and cemented carbide tools).

We also found that many seed particles that cause the formation of nodules were covered with a similar contamination layer. We believe that these weakly bonded particles were formed on the target surface outside of the racetrack and that they were transferred to the substrate surface immediately after starting the deposition process by the self-repulsion effect. We observed such a contamination layer in the BSE plain-view images on the broken nodular defects as well as in SEM images of FIB cross-sections of nodular defects.

Another result of this study is that the surface roughness of the second TiAlN coating is smaller in comparison with the first one. This phenomenon was explained by the planarization effect.

**Author Contributions:** Design of experiments, 3D profilometry, interpretation of experimental results, manuscript writing, P.P.; preparation of specimen for TEM characterization, SEM and FIB analysis, manuscript review, A.D.; SEM analysis, manuscript review, M.Č. and M.P. All authors have read and agreed to the published version of the manuscript.

**Funding:** This work was supported by the Slovenian Research Agency (program P2-0082, project J2-2509). We also acknowledge funding from the European Regional Development Funds (CENN Nanocenter, OP13.1.1.2.02.006).

**Institutional Review Board Statement:** Not applicable.

**Informed Consent Statement:** Not applicable.

**Data Availability Statement:** Not applicable.

**Acknowledgments:** The authors would also like to thank Goran Dražić for TEM analysis and Jožko Fišer for technical assistance.

**Conflicts of Interest:** The authors declare no conflict of interest.

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
