# Peer review of "Contamination of Substrate-Coating Interface Caused by Ion Etching"

_coatings, doi:10.3390/coatings12060846_

Round 1
Reviewer 1 Report
Authors in this paper were focused on the problem of contamination of target surfaces in an industrial deposition system with four magnetron sources. The contamination of the target with the residual products from the etching process can be prevented by a movable shutter located close to the targets, which collects the sputtered species. However, to achieve a reasonable economics of the deposition process, complicated installations (including shielding and shuttering) need to be avoided. In this study, all experiments were performed on samples prepared in the production batches. It is suitable for publication after modification. Listed below are some comments that might be considered beforehand.
- the authors needs to describe the main results in the abstract (numerically express the differences). Introduction is very long. The conclusions must be improved. Please correct.
-
the framework and detailed algorithm for the proposed method should be provided,
-
details of test rig components – (range, accuracy),
- please also report their variances (statistical models),
-
the results were not at all compared with the research of other authors. It is difficult to evaluate the correctness of the experiments and the results without comparison. It is necessary to find as close as possible research oriented in terms of materials and parameters, because in the present form it is only the presentation of the results.
- however, my main concern is the novelty of the work, which the authors have not highlighted well enough. Similar works exist in the literature, so that it is not clear what new the current work brings. Perhaps the authors could make a few conclusions based on their results or otherwise express the importance of the results in a relevant context, instead of just presenting them and stating that they agree with those reported earlier by others.
Author Response
Please find attached the author's repla to the review report.

Reviewer 2 Report
Comments on coatings-1742288
Comment 1: Will the contamination layer at the interface between the substrate
and the coating lead to poor adhesion of the coating?
Comment 2: What is the contribution of using a double TiAlN coating to identify and
determine composition of contamination layer?
Comment 3: How do authors comment on eliminating the contamination layer for the coating?
Comment 4: Figure 11: Where is CrN?
Comment 5: There is spelling error in caption of figure 6(b) “nodulare defects”. It should be corrected as “nodular defects”.
Author Response
Please find attached author's report to the review report.

Round 2
Reviewer 1 Report
Accept in present form